# Factors associated with difficulty of hospital acceptance of patients suspected to have cerebrovascular diseases: A nationwide population-based observational study

**Nobuhiro Sato**[1]*, **Reo Takaku**[2], **Hidenori Higashi**[3], **Alan Kawarai Lefor**[4], **Takashi Shiga**[5]

1 School of Public Health and Preventive Medicine, Monash University, Melbourne, VIC, Australia,
2 Graduate School of Economics, Hitotsubashi University, Kunitachi, Tokyo, Japan, 3 Department of Emergency and Critical Care Medicine, Japanese Red Cross Wakayama Medical Center, Wakayama City, Wakayama, Japan, 4 Department of Surgery, Jichi Medical University, Shimotsuke, Tochigi, Japan, 5 Department of Emergency Medicine, International University of Health and Welfare, Nasushiobara, Tochigi, Japan

* nobuhiro.sato@monash.edu

**Data Availability Statement:** Due to restrictions on the availability of data due to consent agreements for data security as well as IRB approval, data is

## Abstract

Although it is essential to shorten the interval to initial treatment in the care of acute ischemic stroke, some hospitals in Japan reject requests for hospital acceptance from on-scene emergency medical service personnel because of limited resources, which can cause delays in care. We aimed to assess the risk factors for difficulty of hospital acceptance of patients suspected to have cerebrovascular diseases. We conducted a retrospective analysis of the national ambulance records of the Fire and Disaster Management Agency in Japan in 2016. Multivariable logistic regression analysis was used to assess the association between difficulty of hospital acceptance of patients suspected to have cerebrovascular diseases and prehospital factors. During the study period, a total of 222,926 patients were included, and 5283 patients (2.4%) experienced difficulties in hospital acceptance. In multivariable analysis, nights (adjusted odds ratio [AOR] 1.54, 95% confidence interval [CI] 1.45–1.64), weekends (AOR 1.32, 95% CI 1.24–1.40), <25 percentile ratio of emergency physicians and neurosurgeons to all physicians (AOR 1.13, 95% CI 1.03–1.23) (AOR 1.36, 95% CI 1.25–1.48), and mean age of physicians (AOR 1.06, 95% CI 1.05–1.07) were significantly associated with difficulties of hospital acceptance of patients suspected to have cerebrovascular disease. There was a marked regional variation in the difficulties of hospital acceptance. Among the national ambulance records of patients suspected to have cerebrovascular diseases, certain prehospital factors such as weekends were positively associated with difficulty of hospital acceptance. A comprehensive strategy for hospital acceptance of patients with cerebrovascular diseases considering regional variation is required.

available on request. The Japanese government owns the data and interested researchers can contact Ministry of Internal Affairs and Communications Fire and Disaster Management Agency Ambulance Service Planning Office. phone number: +81-3-5253-7529. The data from the Japanese National Survey could be accessed by contacting ministry of health labor and welfare (phone number: +81-3-5253-1111) for researchers who meet the criteria for access to confidential data. The data from Nihon Ultmarc INC could be accessed by contacting https://www.ultmarc.co.jp/mdb/index.html for researchers who meet the criteria for access to confidential data.

**Funding:** The authors received no specific funding for this work.

**Competing interests:** The authors have declared that no competing interests exist.

**Abbreviations:** AOR, adjusted odds ratio; CI, confidence interval; EMS, emergency medical service; FDMA, Fire and Disaster Management Agency; IQR, interquartile range; OR: odds ratio.

# Introduction

The number of ambulance dispatches in Japan increased to 6,340,000 in 2017 [1]. The emergency medical system in Japan is different from that in other countries. The medical staff in charge of a hospital emergency department can decide whether to accept or to reject a request for patient acceptance from the on-scene emergency medical service (EMS) personnel [2, 3]. The person who responds to the phone call (physician, nurse, or other staff) from EMS personnel depends on the hospital. In reality, emergency departments often decline to accept patients because of limited resources, e.g., no available hospital beds or the absence of specialists appropriate for the patient's symptoms [4]. A total of 16.4% of requests for patient acceptance from the on-scene EMS personnel were rejected by medical staff in 2017 [1]. One criterion to define difficulty in hospital acceptance, used by the Fire and Disaster Management Agency (FDMA), is ≥4 phone calls by EMS personnel to hospitals before obtaining acceptance from a destination hospital [2]. The proportion of ≥4 phone calls by EMS personnel to hospitals until acceptance was 2.4% (137,833/5,736,086 patients) in 2017 [1]. An increased number of phone calls to hospitals from ambulances leads to delays in hospital arrival time [2, 3, 5, 6]. As a result, the patient's transport to the hospital may be critically delayed. Prior studies showed that ≥5 phone calls led to more than 16 minutes prolongation compared with one phone call [2, 5, 6].

Stroke is one of the leading causes of death and long-term disability and is the fourth leading cause of death in Japan [7]. As a stroke progresses, neurons are rapidly and irretrievably lost. The typical patient loses 1.9 million neurons each minute that a stroke is untreated [8]. Indeed, in patients with acute ischemic stroke, early administration of intravenous recombinant tissue plasminogen activator within 4.5 hours improves neurological outcomes [9, 10]. When using endovascular therapy, time to revascularization remains the most critical metric for improved clinical outcomes [11, 12]. Every effort should be made to shorten delay in initiation of treatment to improve the outcomes. Therefore, EMS personnel must transport patients suspected to have an acute stroke to hospitals which can manage stroke as soon as possible. Despite their clinical and public health importance, there is a dearth of research to investigate prehospital factors associated with the difficulty of hospital acceptance among patients with stroke in Japan [6].

We hypothesized that patient characteristics such as age, time of day and number of physicians per population would influence hospital acceptance of patients with stroke and there would be regional variations. The aim of this study was to investigate the factor associated with difficulty of hospital acceptance of patients suspected to have cerebrovascular diseases.

# Materials and methods

## Study design and participants

This study analyzed the national ambulance records of the FDMA in Japan from 1 January 2016 to 31 December 2016. These data include all emergency transports throughout Japan except for Tokyo prefecture because fire stations in Tokyo are managed by an organization independent of the national government, and are not included in the FDMA database [13]. The data were collected by EMS personnel, in cooperation with the physicians overseeing the patient's care. This study included all patients ≧15 years of age suspected to have cerebrovascular diseases by physicians at hospitals, who called an ambulance and were transported to a hospital. Hemiplegia, dysarthria, ataxia or severe headache were classified as cerebrovascular disease-related symptoms. We excluded patients with cardio-pulmonary arrest at the hospital and who had missing data. This study was approved by the Ethics Committee of the

International University of Health and Welfare (5-19-46), and the requirement for patient informed consent was waived.

## Setting

Japan has an area of 378,000 km$^2$ divided into 47 prefectures and the population was approximately 127 million in 2016 [14]. The EMS system in Japan has been described elsewhere [15]. There were 733 fire stations with dispatch centers in 2016; EMS at these fire stations is provided by municipal governments [16]. In most cases, an ambulance has a crew of three providers, including at least one emergency lifesaving technician, a person who has undergone extensive training in the provision of pre-hospital care [17, 18]. Using the protocol established by each municipal fire department, EMS ambulance crews at the scene or emergency dispatchers select an appropriate hospital for emergency care according to medical urgency or the patient's symptoms.

## Data collection

Demographic factors (age and gender), chronological factors (date and time), severity, location, and prefecture were extracted from available data. The severity is classified into 4 categories: dead, severe, moderate and mild [13]. Severe patients are those expected to be hospitalized for over 3 weeks, and moderate patients are those expected to be hospitalized for 3 weeks or less. If patients are not likely to require hospitalization, they are categorized as mild. The location where the emergency occurred is also recorded using the following 5 categories: patient's home, public area, workplace, road and others. We defined seven geographic regions (Hokkaido-Tohoku, Kanto, Chubu, Kansai, Chugoku, Shikoku, and Kyushu-Okinawa), based on previous studies [19, 20]. To characterize regional (secondary health care area) level effects in these analyses, we obtained data on the following variables at the prefecture level from the Japanese National Survey as well as Japan national physician database provided by Nihon Ultmarc INC: numbers of physicians, female physicians, emergency physicians, and neurosurgeon, population in the area covered by each municipal fire department, the number of elderly people (≧65 years of age), young people (< 15 years of age) per capita income, industry ratio, and mean age of physicians in each area.

## Outcome measures

The primary outcome was the difficulty of hospital acceptance of patients suspected to have cerebrovascular diseases [6]. The definition of difficulty of hospital acceptance was the requirement for ≥4 phone calls by EMS personnel to hospitals before obtaining acceptance from destination hospitals, based on reports from the FDMA [2]. The secondary outcome was transportation time from arrival at the scene to arrival at the hospital.

## Statistical analysis

Continuous data with skewed distributions was shown as medians and interquartile range (IQR), and categorical data as frequencies and proportions. Bivariate analyses were performed with chi-squared tests for dichotomous variables and the Mann-Whitney U test used for continuous variables. While we use the Mann-Whitney U test to address skewed distribution of some variables, note that the results are almost the same with those obtained using the t-test. For example, the distribution of transportation time is slightly skewed (e.g. skewness = 2.35) but we confirmed that the distribution is sufficiently normal around the mean.

Multivariable analyses were used to assess factors associated with difficulty of hospital acceptance of patients suspected to have cerebrovascular diseases using logistic regression models, and odds ratios (ORs) with 95% confidence intervals (CIs) were calculated. Multivariate linear regression analysis was used to investigate the association between the factors and a reduction in pre-hospital transportation time. We selected covariates based on biological plausibility and previous studies in the multivariable analysis [2, 6, 21]. These variables included age (15–64 years, 65–84 years, ≧85 years), gender (male, female), time of day (daytime [09.00–16.59], night time [17.00–08.59]), day of the week (weekday, weekend), severity (mild, moderate, severe), location (home, public space, workplace, road, others), region (Hokkaido-Tohoku, Kanto, Chubu, Kansai, Chugoku, Shikoku, and Kyushu-Okinawa), month (January, February, March, April, May, June, July, August, September, October, November, December), the number of physicians per population in the area covered by each municipal fire department (<25 percentile, 25–74 percentile, ≧75 percentile), the proportion of emergency physicians to all physicians in the area (<25 percentile, 25–74 percentile, ≧75 percentile), the proportion of neurosurgeon to all physicians in the area (<25 percentile, 25–74 percentile, ≧75 percentile), population in the area, the proportion of older people to all people in the area, the proportion of younger people to all people in the area, per capita income in the area, industry ratio in the area (primary, secondary, tertiary), mean age of physicians in the area, and the proportion of female physicians in the area. In subgroup analyses, we stratified the model according to severity.

Data were analyzed using Stata version 14 (College Station, TX). All tests were two-tailed, and p values <0.05 were statistically significant.

## Results

From January to December 2016, 4,805,224 ambulances were dispatched. Of these, 222,926 patients were eligible for inclusion in this study (Fig 1).

### Patient characteristics

Patient characteristics according to the number of phone calls to hospitals by emergency medical service personnel are listed in Table 1. A total of 5283 (2.4%) patients had difficulty obtaining hospital acceptance. There were differences in the groups regarding age, gender, time, day of the week, severity, location, region, number of physicians per population, proportion of emergency physician and neurosurgeon, and proportion of female physician. The mean age of physicians in the area was similar among the groups. The median transportation time was 36 (IQR 29–45) minutes during the night and 34 (28–43) minutes during daytime (Table 2). The Kanto region had the longest transportation times.

### Risk factors for difficulty of hospital acceptance of patients suspected to have cerebrovascular diseases

In multivariable logistic analysis, a positive association was observed between difficulty of hospital acceptance and night hours (adjusted OR [AOR] 1.54, 95% CI 1.45–1.64), weekend day (AOR 1.32, 95% CI 1.24–1.40), <25 percentile ratio of emergency physicians and neurosurgeons to all physicians (AOR 1.13, 95% CI 1.03–1.23) (AOR 1.36, 95% CI 1.25–1.48), and mean age of physicians in the area (AOR 1.06, 95% CI 1.05–1.07) (Table 3). Conversely, patients 65–84 years of age (AOR 0.86, 95% CI 0.80–0.91), moderate severity (AOR 0.87, 95% CI 0.81–0.94), public space (AOR 0.91, 95% CI 0.85–0.99), and workplace (AOR 0.62, 95% CI 0.51–0.75) were negatively associated with difficulty obtaining hospital acceptance. In addition, there was a high degree of variation in difficulty of hospital acceptance across regions (Fig 2). In multivariate linear regression analysis, similar associations were observed between an increase in pre-hospital

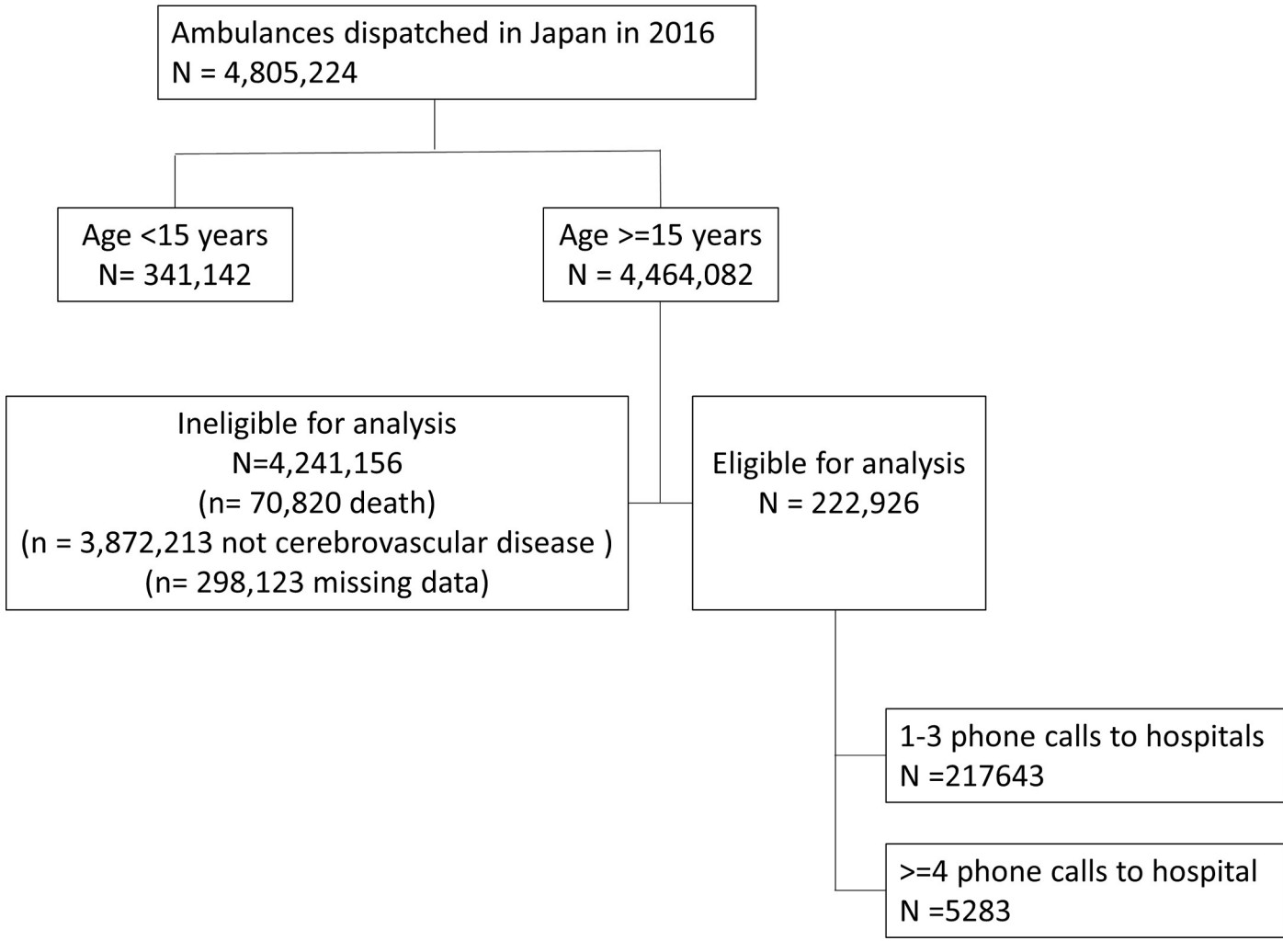

**Fig 1. Study patient flow.**

transportation time and night hours, weekend days, 25–74 percentile ratio of neurosurgeons to all physicians and mean age of physicians in the area, while the associations were observed between a reduction in pre-hospital transportation time and public space, and workplace. In sub-group analysis, similar associations among the above factors except for public space and 25–74 percentile ratio of neurosurgeons to all physicians were observed.

## Discussion

Using data from a national ambulance database in Japan, we observed that nights, weekends, and a higher mean age of physicians in the area are associated with increased difficulty of hospital acceptance and an increased transportation time of patients suspected to have cerebrovascular diseases. These data also demonstrate there is a marked regional variation in the difficulties associated with hospital acceptance.

### Results in context

To our knowledge, this is the first study to document the difficulty of hospital acceptance of patients suspected to have cerebrovascular diseases [6]. A previous study in Osaka, Japan

**Table 1. Basic characteristics and number of phone calls to hospitals by emergency medical service.**

| variables | All number of phone calls to hospitals by EMS | | | | |
| --- | --- | --- | --- | --- | --- |
| | <4 | | ≥4 | | |
| | n = 217643 | % | n = 5283 | % | p value |
| Age, median (IQR) | 76(65–84) | | 75(63–83) | | <0.001* |
| Age | | | | | <0.001† |
| 15–64 | 53,721 | 24.7 | 1,433 | 27.1 | |
| 65–84 | 118,512 | 54.5 | 2,704 | 51.2 | |
| 85- | 50,693 | 23.3 | 1,146 | 21.7 | |
| Gender | | | | | 0.015† |
| Male | 120,143 | 55.2 | 2,937 | 55.6 | |
| Female | 102,783 | 47.2 | 2,346 | 44.4 | |
| Time | | | | | <0.001† |
| Daytime (9:00–16:59) | 84,429 | 38.8 | 1,506 | 28.5 | |
| Night (17:00–8:59) | 138,497 | 63.6 | 3,777 | 71.5 | |
| Day of the week | | | | | <0.001† |
| weekday | 159,841 | 73.4 | 3,468 | 65.6 | |
| weekend | 63,085 | 29.0 | 1,815 | 34.4 | |
| Severity | | | | | <0.001† |
| Minor | 54,302 | 25.0 | 1,202 | 22.8 | |
| Moderate | 126,750 | 58.2 | 2,959 | 56.0 | |
| Severe | 41,874 | 19.2 | 1,122 | 21.2 | |
| Location | | | | | <0.001† |
| Home | 156,742 | 72.0 | 3,848 | 72.8 | |
| Public space | 46,191 | 21.2 | 946 | 17.9 | |
| Workplace | 7,299 | 3.4 | 110 | 2.1 | |
| Road | 7,325 | 3.4 | 176 | 3.3 | |
| Others | 5,370 | 2.5 | 203 | 3.8 | |
| Region | | | | | <0.001† |
| Hokkaido—Tohoku | 30,912 | 14.2 | 838 | 15.9 | |
| Chubu | 48,256 | 22.2 | 1,502 | 28.4 | |
| Kanto | 34,844 | 16.0 | 366 | 6.9 | |
| Kansai | 39,508 | 18.2 | 1,511 | 28.6 | |
| Chugoku | 17,874 | 8.2 | 429 | 8.1 | |
| Shikoku | 11,452 | 5.3 | 227 | 4.3 | |
| Kyusyu, Okinawa | 40,081 | 18.4 | 410 | 7.8 | |
| Month | | | | | |
| January | 21,281 | 9.8 | 985 | 18.6 | |
| February | 18,108 | 8.3 | 604 | 11.4 | |
| March | 19,240 | 8.8 | 552 | 10.5 | |
| April | 17,863 | 8.2 | 393 | 7.4 | |
| May | 17,479 | 8.0 | 393 | 7.4 | |
| June | 16,707 | 7.7 | 299 | 5.7 | |
| July | 16,502 | 7.6 | 317 | 6.0 | |
| August | 16,499 | 7.6 | 292 | 5.5 | |
| September | 17,141 | 7.9 | 321 | 6.1 | |
| October | 18,368 | 8.4 | 342 | 6.5 | |
| November | 18,606 | 8.6 | 372 | 7.0 | |
| December | 19,849 | 9.1 | 413 | 7.8 | |

(*Continued*)

**Table 1.** (Continued)

| variables | All number of phone calls to hospitals by EMS | | | | |
|---|---|---|---|---|---|
| | <4 | | ≥4 | | |
| | n = 217643 | % | n = 5283 | % | p value |
| Number of physicians per population | | | | | <0.001† |
| < 25 percentile | 59,291 | 27.2 | 1,402 | 26.5 | |
| 25–74 percentile | 100,741 | 46.3 | 2,219 | 42.0 | |
| ≧ 75 percentile | 62,893 | 28.9 | 1,662 | 31.5 | |
| Proportion of emergency physicians | | | | | <0.001† |
| < 25 percentile | 63,707 | 29.3 | 1,353 | 25.6 | |
| 25–74 percentile | 105,287 | 48.4 | 2,479 | 46.9 | |
| ≧ 75 percentile | 53,932 | 24.8 | 1,451 | 27.5 | |
| Proportion of neurosurgeons | | | | | <0.001† |
| < 25 percentile | 57,941 | 26.6 | 1,310 | 24.8 | |
| 25–74 percentile | 104,123 | 47.8 | 2,780 | 52.6 | |
| ≧ 75 percentile | 60,862 | 28.0 | 1,193 | 22.6 | |
| Mean age of physicians | 49.94 | | 50.00 | | 0.319* |
| Proportion of female physicians | 20% | | 21% | | <0.001* |
| Transportation time (from arrival at the scene to arrival at the hospital), median | 8.40 | | 8.65 | | <0.001* |

EMS, emergency medical service; IQR, interquartile range

* Mann-Whitney U test

† chi-squared test

reported that elderly patients, foreigners, unconsciousness, nights and weekends/holidays were associated with difficulty of hospital acceptance at the scene requiring EMS personnel to make ≥5 phone calls to hospitals until the patient was accepted for transport [6]. Another study showed, among older patients, more advanced age, nights, weekend days and gastrointestinal related symptoms were more associated with difficulties of hospital acceptance while patients with cardiac arrest, acute coronary syndrome and stroke-related symptoms were less likely to have difficulties of hospital acceptance [2]. The results of the present nationwide study demonstrate a positive association between chronological factors such as nights and weekends and difficulties of hospital acceptance, consistent with previous studies. In Japan, the number of medical facilities and staff that can treat emergency patients during nights, weekends or holidays is low [6]. In addition, few hospitals in Japan have physicians who work in shifts, and they usually continue to work for long hours regardless of the time of day [2]. Furthermore, this study underscored that a higher mean age of physicians in the area is associated with greater difficulty of hospital acceptance of patients suspected to have cerebrovascular diseases. It might be challenging for senior physicians to work longer hours to accept emergency patients without working in shifts. Therefore, centralization of medical resources such as physicians, other specialized staff, and equipment to prevent unbalanced seniority of physicians and longer work hours, using a dedicated emergency physician model of emergency care which allows emergency physicians to work in shifts, might be helpful to facilitate acceptance of the patients during nights or weekends [21].

There was a high degree of variation in difficulties associated with hospital acceptance across regions. In particular, Hokkaido-Tohoku, Chubu, Chugoku, Shikoku, and Kyushu-Okinawa were negatively associated with difficulties of hospital acceptance compared with Kanto and Kansai which include big cities like Yokohama, Osaka and Kyoto. This is consistent with a

**Table 2. Basic characteristics and EMS transportation times.**

| variables | transportation times | |
|---|---|---|
| | median | IQR |
| Age | | |
| 15–64 | 34 | 28–44 |
| 65–84 | 35 | 29–45 |
| 85- | 35 | 28–44 |
| Gender | | |
| Male | 35 | 28–44 |
| Female | 35 | 28–44 |
| Time | | |
| Daytime (9:00–16:59) | 34 | 28–43 |
| Night (17:00–8:59) | 36 | 29–45 |
| Day of the week | | |
| weekday | 35 | 28–44 |
| weekend | 35 | 28–45 |
| Severity | | |
| Minor | 34 | 28–43 |
| Moderate | 35 | 28–45 |
| Severe | 36 | 29–46 |
| Location | | |
| Home | 36 | 29–45 |
| Public space | 33 | 27–41 |
| Workplace | 32 | 26–41 |
| Road | 34 | 27–44 |
| Others | 38 | 31–48 |
| Region | | |
| Hokkaido, Tohoku | 36 | 29–47 |
| Tokai, Chubu, Hokuriku | 33 | 27–42 |
| Kanto | 39 | 32–49 |
| Kansai | 35 | 28–43 |
| Chugoku | 36 | 29–46 |
| Shikoku | 34 | 27–43 |
| Kyusyu, Okinawa | 31 | 25–40 |
| Month | | |
| January | 36 | 29–46 |
| February | 36 | 29–45 |
| March | 35 | 28–44 |
| April | 35 | 28–44 |
| May | 35 | 28–44 |
| June | 35 | 28–44 |
| July | 35 | 28–44 |
| August | 35 | 28–44 |
| September | 35 | 28–44 |
| October | 35 | 28–44 |
| November | 35 | 28–45 |
| December | 35 | 29–45 |
| Number of physicians per population | | |
| < 25 percentile | 37 | 29–46 |

(*Continued*)

**Table 2.** (Continued)

| variables | transportation times | |
| --- | --- | --- |
| | median | IQR |
| 25–74 percentile | 35 | 28–43 |
| ≧ 75 percentile | 32 | 26–40 |
| Proportion of emergency physicians | | |
| < 25 percentile | 37 | 29–47 |
| 25–74 percentile | 36 | 29–46 |
| ≧ 75 percentile | 35 | 28–44 |
| Proportion of neurosurgeons | | |
| < 25 percentile | 36 | 29–46 |
| 25–74 percentile | 35 | 29–44 |
| ≧ 75 percentile | 35 | 28–44 |

EMS, emergency medical service; IQR, interquartile range

prior report which showed that serial rejection of a patient by several hospitals is more frequent in urban areas, where there are many hospitals, but also many patients [4]. To address this issue, a regionalized stroke system which includes establishing primary stroke centers that can deliver intravenous alteplase and better access to those centers is effective [22, 23]. In addition, a law which punishes hospitals if a patient is declined entry to an emergency department for screening and stabilization might be useful, similar to the Emergency Medical Treatment and Active Labor Act (EMTALA) in the United States [24].

## Limitations

The present study has several acknowledged limitations. First, we were unable to obtain information regarding in-hospital outcomes and treatment of patients after arrival at the hospital. Therefore, we did not classify cerebrovascular diseases as acute ischemic stroke or cerebral hemorrhage. Second, the actual severity of a patient's condition might not be reflected by the severity judged at the time when the ambulance was called, because further assessment of the patient's condition was conducted upon arrival at the hospital. Some patients might deteriorate during transport and be assessed as critically ill on arrival at the hospital. Third, we could not analyze information regarding consciousness, such as Glasgow Coma Scale or Japan Coma Scale because the data were not available. Fourth, we did not consider national holidays, which were included as weekdays. Finally, despite adjusting for potential covariates, we did not exclude other possible residual confounding factors that might affect difficulties associated with hospital acceptance of patients suspected to have cerebrovascular diseases, such as the area's hospital bed capacity or occupancy rates.

## Conclusion

The results of this study show that prehospital factors of nights, weekends, and a higher mean age of physicians in the area are associated with greater difficulty of hospital acceptance and increased transportation time for patients suspected to have cerebrovascular diseases and that there is a marked regional variation. A comprehensive strategy to facilitate hospital acceptance of patients suspected to have cerebrovascular diseases considering regional variation is required.

**Table 3. Factors associated with difficulty of hospital acceptance and transportation time among patients suspected to have cerebrovascular diseases.**

| | All patients | | | |
| --- | --- | --- | --- | --- |
| | Number of phone calls to hospitals by EMS | | Transportation time | |
| | OR | 95% CI | Time Difference | 95% CI |
| Age | | | | |
| 15–64 | reference | | reference | |
| 65–84 | 0.86*** | 0.80–0.91 | -0.17 | -0.44–0.09 |
| 85- | 0.93* | 0.86–1.01 | -0.60*** | -0.96 - -0.24 |
| Female | 0.96 | 0.91–1.02 | -0.29*** | -0.39 - -0.19 |
| Night | 1.54*** | 1.45–1.64 | 1.31*** | 1.03–1.58 |
| Weekend | 1.32*** | 1.24–1.40 | 0.67*** | 0.41–0.92 |
| Severity | | | | |
| Mild | reference | | reference | |
| Moderate | 0.87*** | 0.81–0.94 | 0.32 | -0.26–0.91 |
| Severe | 0.92** | 0.84–1.00 | 0.53 | -0.30–1.35 |
| Location | | | | |
| Home | reference | | reference | |
| Public space | 0.91** | 0.85–0.99 | -1.31*** | -1.66 - -0.97 |
| Workplace | 0.62*** | 0.51–0.75 | -2.35*** | -2.70 - -2.00 |
| Road | 1.00 | 0.85–1.16 | -0.93*** | -1.38 - -0.49 |
| Others | 1.45*** | 1.25–1.70 | 0.78 | -0.87–2.44 |
| Region | | | | |
| Hokkaido—Tohoku | 0.69*** | 0.61–0.77 | -3.58 | -8.37–1.21 |
| Chubu | 0.35*** | 0.31–0.40 | -5.69** | -10.33 - -1.04 |
| Kanto | reference | | reference | |
| Kansai | 1.10* | 1.00–1.20 | -4.00** | -7.83 - -0.16 |
| Chugoku | 0.81*** | 0.71–0.92 | -3.94 | -8.97–1.09 |
| Shikoku | 0.74*** | 0.63–0.87 | -6.57*** | -11.30 - -1.83 |
| Kyusyu, Okinawa | 0.22*** | 0.19–0.26 | -8.39*** | -13.72 - -3.05 |
| Month | | | | |
| January | reference | | reference | |
| February | 0.73*** | 0.65–0.81 | -0.34* | -0.71–0.03 |
| March | 0.62*** | 0.56–0.69 | -0.91*** | -1.27 - -0.55 |
| April | 0.47*** | 0.42–0.53 | -1.11*** | -1.53 - -0.69 |
| May | 0.47*** | 0.42–0.53 | -1.30*** | -1.74 - -0.85 |
| June | 0.38*** | 0.33–0.43 | -1.60*** | -2.10 - -1.11 |
| July | 0.40*** | 0.36–0.46 | -1.75*** | -2.35 - -1.15 |
| August | 0.37*** | 0.32–0.42 | -1.73*** | -2.25 - -1.20 |
| September | 0.40*** | 0.35–0.45 | -1.41*** | -1.98 - -0.83 |
| October | 0.39*** | 0.35–0.45 | -1.36*** | -1.92 - -0.79 |
| November | 0.43*** | 0.38–0.48 | -1.06*** | -1.63 - -0.49 |
| December | 0.45*** | 0.40–0.51 | -1.03*** | -1.54 - -0.52 |
| Number of physicians per population | | | | |
| < 25 percentile | 1.09 | 0.95–1.24 | 2.81* | -0.16–5.79 |
| 25–74 percentile | 0.95 | 0.85–1.04 | 1.33 | -0.91–3.56 |
| ≧ 75 percentile | reference | | reference | |
| Proportion of emergency physicians | | | | |
| < 25 percentile | 1.13*** | 1.03–1.23 | 0.86 | -0.37–2.08 |
| 25–74 percentile | 1.17*** | 1.08–1.26 | 0.657 | -0.87–2.18 |

*(Continued)*

**Table 3.** (Continued)

| | All patients | | | |
|---|---|---|---|---|
| | Number of phone calls to hospitals by EMS | | Transportation time | |
| | OR | 95% CI | Time Difference | 95% CI |
| ≧ 75 percentile | reference | | reference | |
| Proportion of neurosurgeons | | | | |
| < 25 percentile | 1.36*** | 1.25–1.48 | 1.362** | 0.10–2.61 |
| 25–74 percentile | 1.32*** | 1.22–1.44 | 1.432*** | 0.46–2.39 |
| ≧ 75 percentile | reference | | reference | |
| Mean age of physicians | 1.06*** | 1.05–1.07 | 0.29*** | 0.09–0.48 |
| Proportion of female physicians | 1.38 | 0.76–2.53 | -2.24 | -10.28–5.80 |

EMS, emergency medical service; OR, odds ratio; CI, confidence interval

* p < 0.10

** p < 0.05

*** p < 0.01

Nagelkerke pseudo R2 = 0.053

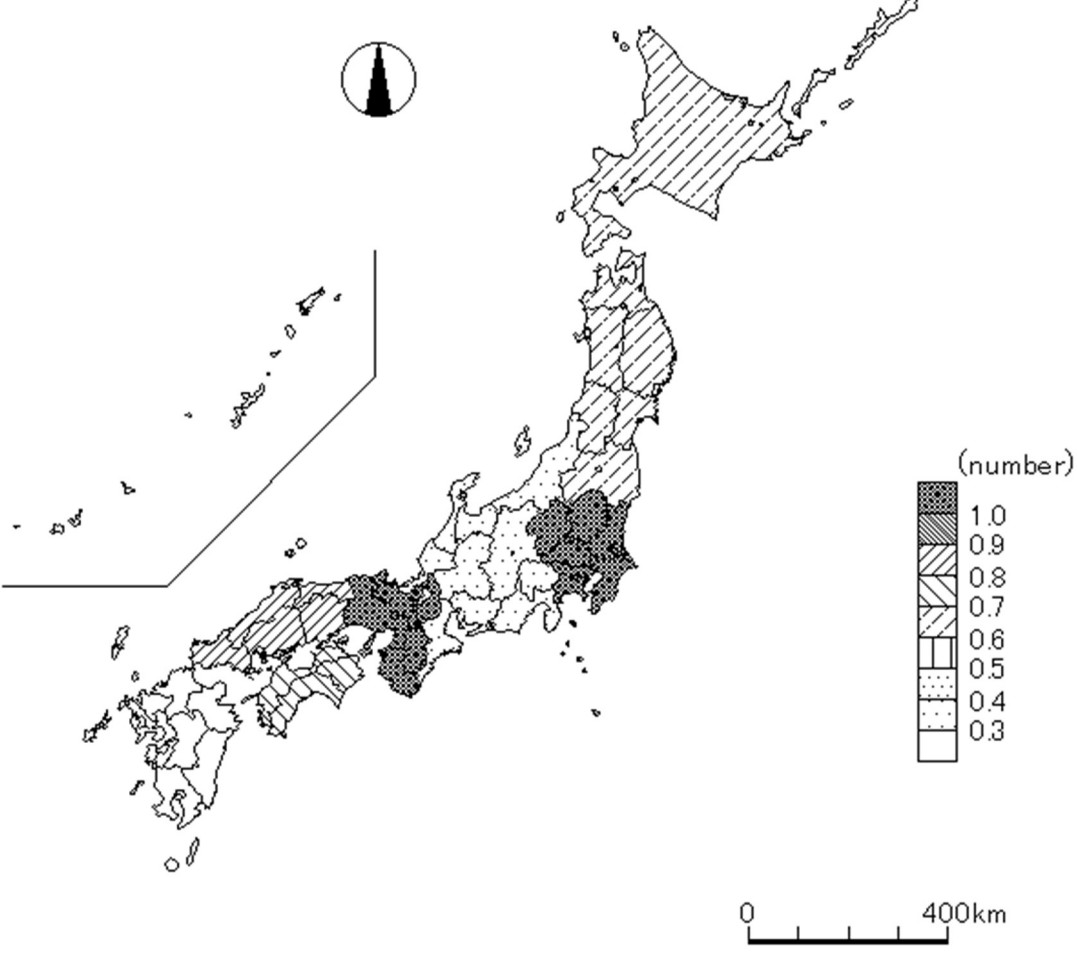

**Fig 2. Regional variation of the odds ratio of number of phone calls to hospitals by emergency medical service personnel.**

## Acknowledgments

We would like to thank all of the emergency medical services personnel.

## Author Contributions

**Conceptualization:** Nobuhiro Sato, Reo Takaku, Hidenori Higashi, Takashi Shiga.

**Data curation:** Reo Takaku.

**Formal analysis:** Reo Takaku.

**Investigation:** Nobuhiro Sato, Takashi Shiga.

**Methodology:** Nobuhiro Sato, Reo Takaku, Hidenori Higashi, Takashi Shiga.

**Project administration:** Nobuhiro Sato, Takashi Shiga.

**Supervision:** Nobuhiro Sato, Takashi Shiga.

**Visualization:** Nobuhiro Sato, Takashi Shiga.

**Writing – original draft:** Nobuhiro Sato.

**Writing – review & editing:** Alan Kawarai Lefor, Takashi Shiga.

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
