## [Decision Letter · Decision Letter 0]

18 May 2020

PONE-D-20-03553

Factors associated with difficulty of hospital acceptance of patients suspected to have cerebrovascular diseases: A nationwide population-based observational study

PLOS ONE

Dear Dr Sato,

Thank you for submitting your manuscript to PLOS ONE. After careful consideration, we feel that it has merit but does not fully meet PLOS ONE’s publication criteria as it currently stands. Therefore, we invite you to submit a revised version of the manuscript that addresses the points raised during the review process.

We would appreciate receiving your revised manuscript by Jul 02 2020 11:59PM. To enhance the reproducibility of your results, we recommend that if applicable you deposit your laboratory protocols in protocols.io, where a protocol can be assigned its own identifier (DOI) such that it can be cited independently in the future. For instructions see: http://journals.plos.org/plosone/s/submission-guidelines#loc-laboratory-protocols

We look forward to receiving your revised manuscript.

Kind regards,

Ho Ting Wong, PhD

Academic Editor

PLOS ONE

Journal Requirements:

Additional Editor Comments (if provided):

I think you can add more content to the introduction section.

For table 2, please provide p-values, Nagelkerke pseudo R2, and the information on how you code the variables.

Why are some variables, such as “month,” missing from the table?

In line 162, you mentioned that the variable “month” was considered in the regression analysis.

Reviewers' comments:

Reviewer's Responses to Questions

**Comments to the Author**

1. Is the manuscript technically sound, and do the data support the conclusions?

Reviewer #1: Yes

Reviewer #2: No

2. Has the statistical analysis been performed appropriately and rigorously? 

Reviewer #1: Yes

Reviewer #2: No

3. Have the authors made all data underlying the findings in their manuscript fully available?

Reviewer #1: Yes

Reviewer #2: No

4. Is the manuscript presented in an intelligible fashion and written in standard English?

Reviewer #1: Yes

Reviewer #2: Yes

5. Review Comments to the Author

Reviewer #1: In the present study, the authors aimed to investigate the factor associated with difficulty of hospital acceptance of patients suspected to have cerebrovascular diseases. When reviewing this manuscript, I am worrying that hospitals in Japan should be increasingly turning away sick people as the country struggles with surging coronavirus infections and its emergency medical system collapses.

Overall, this manuscript is well written. I only have the following minor comments:

1. In Line 103, “we excluded patients who were children (< 15 years old)…” And in Line 136, they said “…young people (< 15 years of age).” I suggest the authors to use a consistent term for < 15 years of age.

2. Line 124: In addition to the 4 categories of severity, can the authors accessed other variables (e.g. Japan Coma Scale) in their data set?

3. Line 129-130, they defined seven geographic regions (Hokkaido-Tohoku, Kanto, Chubu, Kansai, Chugoku, Shikoku, and Kyushu-Okinawa). But In Table 1, the name of the seven regions did not 100% match those written in Line 129-130. It’s better to revise it.

4. Line 164-168: “…the number of physicians per population in the area covered by each municipal fire department (<25 percentile, 25-75 percentile, 75 percentile≦), the proportion of emergency physicians to all physicians in the area (<25 percentile, 25-75 percentile, 75 percentile≦), the proportion of neurosurgeon to all physicians in the area (<25 percentile, 25-75 percentile, 75 percentile≦),…” Should “75 percentile≦” better be “76 percentile≦”? That’s because “25-75 percentile” already included 75.

5. As above, the use of “25-75 percentile” as reference was less intuitive and would made the results difficult to interpret. I suggest to use “76 percentile≦” as a reference.

6. Line 262: May add a limitation that the authors did not include national holidays, in addition to weekend, into analysis.

Keep fingers crossed and hope that Japan, as well as the whole world, will get through the tough situation right now.

Reviewer #2: This study expanded the scope of prior research on "Factors associated with the difficulty in hospital acceptance at the scene by emergency medical service personnel" and “Evaluation of factors associated with the difficulty in finding receiving hospitals for traffic accident patients” by examining similar factors associated with nationwide ambulance diversion or so-called “difficulty of hospital acceptance.” The outcome is measured as four or more phone calls to destination hospitals prior to the arrival. The topic is important given the high prevalence of patients being declined for emergency medical services in Japan; understanding the prehospital factors associated with ambulance diversion may help identify at-risk vulnerable populations and further provide policy measures to address the difficulties. Several concerns dampen my enthusiasm for the overall merit of the paper. I hope my comments and suggestions below help the authors as they move forth with this research.

• Lack of contribution: Authors did a great job discussing the importance of timely access to treatments among patients with stroke. However, besides the essential role of EMS and timely hospital acceptance on stroke care, this paper would benefit from a brief discussion on why and how authors would hypothesize different findings of prehospital factors associated with the difficulty in hospital acceptance between trauma care and stroke care. In other words, what makes stroke patients special in terms of prehospital factors for hospital acceptance. The findings are mostly consistent with previous studies, making the contribution of this new article unclear.

• the term, “difficulty of hospital acceptance”: is it different from ambulance diversion? the noun phrase difficulty seems to be missing a determiner before it. Authors might consider adding an article before ‘difficulty’ throughout the paper.

• Hospital bed capacity as a confounder: surprisingly, authors noted hospital bed capacity as one key reason for the difficulty in hospital ER acceptance but did not examine this factor in the study. An area’s hospital bed capacity or occupancy rates would be a confounder for the relationship between provider supply and outcomes.

• Page 5, Line 84: “early administration of intravenous recombinant tissue plasminogen activator within 4.5 hours” is interesting, as the study found a minimal difference in median transportation time (<1 minute) between patients in the group of less than 4 phone calls to be accepted by a hospital and their counterparts with 4+ phone calls.

• Descriptive statistics for transportation times by prehospital factors would help readers interpret the model coefficients.

• Student’s t-test statistics should not be used for skewed data even though those are continuous variables. Consider using Wilcoxon tests for the comparisons between medians of the two groups.

• Regression analysis of transportation times: authors might want to clarify whether any clustering effects were controlled in the multivariate logistic and linear regression analyses, given the nature of multiple ecological levels of factors (patient, area, and region). Also, how many patients had multiple ambulance services during the study period?

• Nonlinear associations for transportation times by factors examined: since authors report medians of transportation times in Table 2, it is reasonable to expect a skewed distribution of this outcome. How did authors address the violation of homoscedasticity assumptions and normal distributions of the error terms in the multivariate linear model?

Minor comments:

• Page 5, Line 71: authors stated that “in reality, emergency departments often decline to accept patients because of limited resources.” this would have been a stronger statement if authors can quantify how often this phenomenon was in 2017.

• Page 5, Line 77: the unit of the rate is needed here. Was it 137,833 per 5,736,086 patients?

• Page 5, Line 83: ‘among’ patients with acute ischemic stroke

• Page 6, Line 90-93: these two sentences are a bit repetitive. Authors might consider reconcile these.

• Page 6, Line 101: why was age at 15 chosen as a cutoff?

• Page 7, Line 103-105: the inclusion and exclusion criteria read redundant. Authors may first describe inclusion criteria and state that they also exclude those with cardio-pulmonary arrest at the hospital and who had missing data on which factors.

• Page 7, Line 117-118, “an appropriate hospital” requires an explanation. It’d be great if authors can discuss the protocol of the hospital selections. Were those by proximity from an onset location to a hospital? by hospital bed capacity? or other criteria?

• Page 8, Line 150: “univariate” should be bivariate.

• Table 1 and Table 2: the third quartiles were listed as equal or greater than 75 percentile but 75 percentile has been included in the second categories.

• Table 1: authors may consider presenting both median and interquartiles for transportation times. This may also apply for the new table with the statistics of transportation times by factors.

• Page 14, Line 207-208: it is unclear what the legend in Figure 2 means. why not the predicted rate after controlling for patient and area factors?

• Pape 14, Line 209-214: this sentence was written with ambiguity. Authors stated that similar associations were observed between a “reduction” in prehospital transportation time and night hours, weekend days,… but the table presents opposite for some variables.

• Page 18, Line 258-259: rather than saying the present findings should encourage healthcare providers and policy makers to decrease these regional variations, authors might use existing evidence to support how interventions should be implemented to address the difficulty facing hospital acceptance. Several examples would be a stroke care regionalization, and EMTALA law in the US.

6. PLOS authors have the option to publish the peer review history of their article (what does this mean?). If published, this will include your full peer review and any attached files.

Reviewer #1: Yes: Cheng-Yang Hsieh

Reviewer #2: No

---

## [Author Response · Author response to Decision Letter 0]

9 Jul 2020

Re: PONE-D-20-03553 

Dear Editors and Reviewers:

Thank you for your thorough review of PONE-D-20-03553, “Factors associated with difficulty of hospital acceptance of patients suspected to have cerebrovascular diseases: a nationwide population-based observational study”. Please find our responses to the Editors’ and Reviewers’ comments shown in Bold in the following response letter.

Sincerely,

Nobuhiro Sato

Response to the Editor:

Thank you for your thorough reviews and suggestions. Our responses to your queries follow.

Additional Editor Comments:

I think you can add more content to the introduction section.

We appreciate your suggestions. As suggested, we have added more content to the Introduction section (Page 5-6 Para 65-98). 

For table 2, please provide p-values, Nagelkerke pseudo R2, and the information on how you code the variables.

As requested, we have added information to table 3 and added a new table as table 2.

Why are some variables, such as “month,” missing from the table?

In line 162, you mentioned that the variable “month” was considered in the regression analysis.

We thank the reviewer for the careful review of the manuscript. We have added this information to table 3 (we have made a new table as table 2) as above.

 

Response to Reviewer 1:

Reviewer #1: In the present study, the authors aimed to investigate the factor associated with difficulty of hospital acceptance of patients suspected to have cerebrovascular diseases. When reviewing this manuscript, I am worrying that hospitals in Japan should be increasingly turning away sick people as the country struggles with surging coronavirus infections and its emergency medical system collapses.

Overall, this manuscript is well written. I only have the following minor comments:

Thank you for your thorough reviews and suggestions. Our responses to your queries follow.

1. In Line 103, “we excluded patients who were children (< 15 years old)…” And in Line 136, they said “…young people (< 15 years of age).” I suggest the authors to use a consistent term for < 15 years of age.

We appreciate your suggestions. As pointed by reviewer 2 as well, we have removed <15 years old.

2. Line 124: In addition to the 4 categories of severity, can the authors accessed other variables (e.g. Japan Coma Scale) in their data set?

Unfortunately, the database we used did not include the Japan Coma Scale or Glasgow Coma Scale. Therefore, we have acknowledged this issue in the Limitations (Page 23 Para 282-284).

3. Line 129-130, they defined seven geographic regions (Hokkaido-Tohoku, Kanto, Chubu, Kansai, Chugoku, Shikoku, and Kyushu-Okinawa). But In Table 1, the name of the seven regions did not 100% match those written in Line 129-130. It’s better to revise it.

We thank the reviewer for the careful review. We have revised Table 1 as you suggested.

4. Line 164-168: “…the number of physicians per population in the area covered by each municipal fire department (<25 percentile, 25-75 percentile, 75 percentile≦), the proportion of emergency physicians to all physicians in the area (<25 percentile, 25-75 percentile, 75 percentile≦), the proportion of neurosurgeon to all physicians in the area (<25 percentile, 25-75 percentile, 75 percentile≦),…” Should “75 percentile≦” better be “76 percentile≦”? That’s because “25-75 percentile” already included 75.

We meant 25-75 percentile as 25≦ and <75. Therefore, we have revised to 25-74 percentile in the Materials and Methods, table 1 and table 3 (we have made a new table as table 2) (Page 10-11 Para 166-180).

5. As above, the use of “25-75 percentile” as reference was less intuitive and would made the results difficult to interpret. I suggest to use “76 percentile≦” as a reference.

As requested, we have used 75 percentile≦ as a reference in table 3 (we have made a new table as table 2) .

6. Line 262: May add a limitation that the authors did not include national holidays, in addition to weekend, into analysis.

We appreciate your suggestions. As suggested, we have added the issue to the Limitation section (Page 23 Para 284-285).

Keep fingers crossed and hope that Japan, as well as the whole world, will get through the tough situation right now.

We appreciate your help.

 

Response to Reviewer 2:

Reviewer #2: This study expanded the scope of prior research on "Factors associated with the difficulty in hospital acceptance at the scene by emergency medical service personnel" and “Evaluation of factors associated with the difficulty in finding receiving hospitals for traffic accident patients” by examining similar factors associated with nationwide ambulance diversion or so-called “difficulty of hospital acceptance.” The outcome is measured as four or more phone calls to destination hospitals prior to the arrival. The topic is important given the high prevalence of patients being declined for emergency medical services in Japan; understanding the prehospital factors associated with ambulance diversion may help identify at-risk vulnerable populations and further provide policy measures to address the difficulties. Several concerns dampen my enthusiasm for the overall merit of the paper. I hope my comments and suggestions below help the authors as they move forth with this research.

Thank you for your thorough review and suggestions. Our responses to your queries follow.

• Lack of contribution: Authors did a great job discussing the importance of timely access to treatments among patients with stroke. However, besides the essential role of EMS and timely hospital acceptance on stroke care, this paper would benefit from a brief discussion on why and how authors would hypothesize different findings of prehospital factors associated with the difficulty in hospital acceptance between trauma care and stroke care. In other words, what makes stroke patients special in terms of prehospital factors for hospital acceptance. The findings are mostly consistent with previous studies, making the contribution of this new article unclear.

We thank the reviewer for these insightful comments. We considered that delay of transportation to the hospital for patients with acute stroke was an important problem in public health and clinical situation. One of the main reasons is that stroke was fourth leading cause of death (7.9%) in 2018 in Japan, while the death ratio of accidents from leading causes of death was 3.0% (1). Another reason is that time is brain (2). Human nervous tissue is rapidly and irretrievably lost as stroke progresses. The typical patient loses 1.9 million neurons each minute in which stroke is untreated. In addition, as shown in the Materials and methods, we used nationwide database and characterized regional level effects unlike the previous studies which we believe is a new contribution of this article. We have revised the Introduction (Page 5-6 Para 65-98).

• the term, “difficulty of hospital acceptance”: is it different from ambulance diversion? the noun phrase difficulty seems to be missing a determiner before it. Authors might consider adding an article before ‘difficulty’ throughout the paper

We appreciate the author's insightful comments. Ambulance diversion is often used to describe the practice of temporarily closing a facility, typically an emergency department, to incoming ambulances in the United states (3). In Japan, an emergency department can choose to accept an incoming ambulance on a single case basis. Therefore, we have used the word “difficulty of hospital acceptance” instead of ambulance diversion. We have inserted a reference article throughout the paper per the reviewer's suggestion.

• Hospital bed capacity as a confounder: surprisingly, authors noted hospital bed capacity as one key reason for the difficulty in hospital ER acceptance but did not examine this factor in the study. An area’s hospital bed capacity or occupancy rates would be a confounder for the relationship between provider supply and outcomes

Unfortunately, we did not have the data of an area’s hospital bed capacity or occupancy rates. Therefore, we have added the issue in the Limitation section (Page 23 Para 285-288).

• Page 5, Line 84: “early administration of intravenous recombinant tissue plasminogen activator within 4.5 hours” is interesting, as the study found a minimal difference in median transportation time (<1 minute) between patients in the group of less than 4 phone calls to be accepted by a hospital and their counterparts with 4+ phone calls.

• Descriptive statistics for transportation times by prehospital factors would help readers interpret the model coefficients.

We appreciate these helpful suggestions. As suggested, we have made a new table of descriptive statistics for transportation times by prehospital factors.

• Student’s t-test statistics should not be used for skewed data even though those are continuous variables. Consider using Wilcoxon tests for the comparisons between medians of the two groups.

We appreciate reviewer's insightful comments. We have used Mann-Whitney U test for skewed data for comparisons between medians of the two groups because two groups were independent. Therefore, we have revised Table 1 and Material and Methods (Page 9-10 Para153-159).

• Regression analysis of transportation times: authors might want to clarify whether any clustering effects were controlled in the multivariate logistic and linear regression analyses, given the nature of multiple ecological levels of factors (patient, area, and region). Also, how many patients had multiple ambulance services during the study period?

We appreciate the author's insightful comments. As shown in the Materials and methods, we adjusted factors of each patient and region with the national ambulance records and factors in each area with data at the prefecture level from the Japanese National Survey as well as Japan national physician database. 

Unfortunately, the national data used in the study does not have personal identification information. It has information about each ambulance transport. Therefore, it is not possible to examine the effects of repeated transports of the same patients.

• Nonlinear associations for transportation times by factors examined: since authors report medians of transportation times in Table 2, it is reasonable to expect a skewed distribution of this outcome. How did authors address the violation of homoscedasticity assumptions and normal distributions of the error terms in the multivariate linear model?

We appreciate reviewer's insightful comments. The distribution of transportation time somewhat skewed as you pointed out, but it seems sufficiently normal around the mean. Also, it is well known that the normality assumption for linear regression applies to the errors, not to the outcome variable per se. Since we adjust many covariates, the error terms would be still normal even if the outcome variable exhibits a bit skewness. Anyway, we follow conventional wisdom that t tests will still provide good approximations if the distribution is not too grossly non-normal (and this is actually true for our outcome variable).

Minor comments:

• Page 5, Line 71: authors stated that “in reality, emergency departments often decline to accept patients because of limited resources.” this would have been a stronger statement if authors can quantify how often this phenomenon was in 2017.

16.4% of requests for patient acceptance from the on-scene EMS personnel was rejected by the medical staff once at least in 2017 (4). As suggested, we have added the information in the Introduction (Page 5 Para 71-72).

• Page 5, Line 77: the unit of the rate is needed here. Was it 137,833 per 5,736,086 patients?

As suggested, we have added the unit of the rate (Page 5 Para 75-76).

• Page 5, Line 83: ‘among’ patients with acute ischemic stroke

The prior study was about patients with acute myocardial infarction (5). There was no previous study of association between phone calls and transportation time among patients with acute ischemic stroke. We have revised it (Page 5 Para 79-80).

• Page 6, Line 90-93: these two sentences are a bit repetitive. Authors might consider reconcile these.

We thank the reviewer for the careful review. As suggested, we have revised them.

• Page 6, Line 101: why was age at 15 chosen as a cutoff?

In Japanese medical system, patients who are < 15 years of age are regarded as children while those who are ≧15 years of age are regarded as adults (6). Therefore, we chose age at 15 as a cutoff.

• Page 7, Line 103-105: the inclusion and exclusion criteria read redundant. Authors may first describe inclusion criteria and state that they also exclude those with cardio-pulmonary arrest at the hospital and who had missing data on which factors.

As requested, we have revised it (Page 7 Para 108-109).

• Page 7, Line 117-118, “an appropriate hospital” requires an explanation. It’d be great if authors can discuss the protocol of the hospital selections. Were those by proximity from an onset location to a hospital? by hospital bed capacity? or other criteria?

As shown in the Materials and methods, each municipal fire department has their own medical protocol, which helps emergency medical service decide to which hospital they transfer patients. For example, a stroke patient will be taken to the nearest designated stroke center. A severe trauma patient will be transported to the regional tertiary trauma center by aeromedical transport per the protocol. 

• Page 8, Line 150: “univariate” should be bivariate.

As suggested, we have revised it (Page 9-10 Para 153-155).

• Table 1 and Table 2: the third quartiles were listed as equal or greater than 75 percentile but 75 percentile has been included in the second categories.

We thank the reviewer for the careful review. We have revised Table 1 and Table 3 (we have added a new table 2).

• Table 1: authors may consider presenting both median and interquartiles for transportation times. This may also apply for the new table with the statistics of transportation times by factors.

As requested, we have made the new table with the statistics of transportation times by factors.

• Page 14, Line 207-208: it is unclear what the legend in Figure 2 means. why not the predicted rate after controlling for patient and area factors?

We appreciate the author's insightful comments. Figure 2 was made from the actual rate of each region. Therefore, we used the odds ratio of number of phone calls to hospitals by emergency medical service personnel instead of predicted rate.

• Pape 14, Line 209-214: this sentence was written with ambiguity. Authors stated that similar associations were observed between a “reduction” in prehospital transportation time and night hours, weekend days,… but the table presents opposite for some variables.

We thank the reviewer for these insightful comments. As suggested, we have revised it (Page 17 Para 218-222).

• Page 18, Line 258-259: rather than saying the present findings should encourage healthcare providers and policy makers to decrease these regional variations, authors might use existing evidence to support how interventions should be implemented to address the difficulty facing hospital acceptance. Several examples would be a stroke care regionalization, and EMTALA law in the US.

We appreciate your suggestions. As suggested, we have added existing evidence to address the difficulty of hospital acceptance (Page 22 Para 267-272).

Reference

1. Ministry of Health Labour, and Welfare. Vital Statistics 2018. https://www.mhlw.go.jp/english/database/db-hw/dl/81-1a2en.pdf

2. Saver JL. Time is brain--quantified. Stroke. 2006;37(1):263-6.

3. Geiderman JM, Marco CA, Moskop JC, Adams J, Derse AR. Ethics of ambulance diversion. Am J Emerg Med. 2015;33(6):822-7.

4. Ambulance Service Planning Office of Fire and Disaster Management Agency of Japan: 2018 Effect of first aid for emergency patients. https://www.fdma.go.jp/publication/rescue/items/kkkg_h30_01_kyukyu.pdf.

5. Kitamura T, Iwami T, Kawamura T, Nishiyama C, Sakai T, Tanigawa-Sugihara K, et al. Ambulance calls and prehospital transportation time of emergency patients with cardiovascular events in Osaka City. Acute Med Surg. 2014;1(3):135-44.

6. Katayama Y, Kitamura T, Kiyohara K, Iwami T, Kawamura T, Hayashida S, et al. Evaluation of factors associated with the difficulty in finding receiving hospitals for traffic accident patients at the scene treated by emergency medical services: a population-based study in Osaka City, Japan. Acute Med Surg. 2017;4(4):401-7.

 

Journal Requirements:

 As suggested, we have revised it.

Data Availability Statement: The database is available to researchers approved by the Japanese government through multiple processes to ensure data security, consistent with the laws of Japan. In addition, there are restrictions on the availability of data due to consent agreements for data security as well as IRB approval, which allow access only to external researchers for research monitoring purposes. The Japanese government owns the data and interested researchers can contact Ministry of Internal Affairs and Communications Fire and Disaster Management Agency Ambulance Service Planning Office. phone number: +81-3-5253-7529. 

We used data at the prefecture level from the Japanese National Survey as

well as Japan national physician database provided by Nihon Ultmarc INC.

The data from the Japanese National Survey could be accessed by contacting 

ministry of health labor and welfare (phone number: +81-3-5253-1111) for researchers who meet the criteria for access to confidential data. The data from Nihon Ultmarc INC could be accessed by contacting https://www.ultmarc.co.jp/mdb/index.html for researchers who meet the criteria for access to confidential data.

---

## [Decision Letter · Decision Letter 1]

24 Nov 2020

PONE-D-20-03553R1

Factors associated with difficulty of hospital acceptance of patients suspected to have cerebrovascular diseases: a nationwide population-based observational study

PLOS ONE

Dear Dr. Sato,

Thank you for submitting your manuscript to PLOS ONE. After careful consideration, we feel that it has merit but does not fully meet PLOS ONE’s publication criteria as it currently stands. Therefore, we invite you to submit a revised version of the manuscript that addresses the points raised during the review process.

We look forward to receiving your revised manuscript.

Kind regards,

Ho Ting Wong, PhD

Academic Editor

PLOS ONE

Reviewers' comments:

Reviewer's Responses to Questions

**Comments to the Author**

1. If the authors have adequately addressed your comments raised in a previous round of review and you feel that this manuscript is now acceptable for publication, you may indicate that here to bypass the “Comments to the Author” section, enter your conflict of interest statement in the “Confidential to Editor” section, and submit your "Accept" recommendation.

Reviewer #1: All comments have been addressed

Reviewer #3: All comments have been addressed

2. Is the manuscript technically sound, and do the data support the conclusions?

Reviewer #1: Yes

Reviewer #3: Yes

3. Has the statistical analysis been performed appropriately and rigorously? 

Reviewer #1: Yes

Reviewer #3: Yes

4. Have the authors made all data underlying the findings in their manuscript fully available?

Reviewer #1: Yes

Reviewer #3: Yes

5. Is the manuscript presented in an intelligible fashion and written in standard English?

Reviewer #1: Yes

Reviewer #3: No

6. Review Comments to the Author

Reviewer #1: The authors have addressed all my comments adequately.

The manuscript now looks nice.

I have no further comments.

Reviewer #3: This study aimed to examine the predictors of difficulty of hospital acceptance of patients suspected to have cerebrovascular disease. It is unclear who and how to define those who were suspected to have cerebrovascular diseases? By the EMS personnel? Difficulty of acceptance was defined by having ≥4 phone calls by the EMS personnel to hospitals until acceptance. Who are the one responding to the phone calls? Triage nurses? Hospital administrators? Physicians? Symptom presentation is a key factor to urge patients to call for help, same as healthcare professionals. Did the EMS personnel inform the one responding to the phone calls the symptoms of patients? Did they know that the patients are suspected to have cerebrovascular diseases? Mean age of physicians is one of the factor associated with difficulty of hospital acceptance, was this variable refer to those who refused to admit the patients or those who finally admitted the patients? did these physicians involved in refusing to admit patients?

Minimal grammatical errors, but the manuscript requires some formatting to make it more readable, such as the “≦” should put in front of 75 percentile, Table 1, the fourth column heading should be ≥4

7. PLOS authors have the option to publish the peer review history of their article (what does this mean?). If published, this will include your full peer review and any attached files.

Reviewer #1: No

Reviewer #3: No

---

## [Author Response · Author response to Decision Letter 1]

18 Dec 2020

Re: PONE-D-20-03553R1 

Dear Editors and Reviewers:

Thank you for your thorough review of PONE-D-20-03553R 1, “Factors associated with difficulty of hospital acceptance of patients suspected to have cerebrovascular diseases: a nationwide population-based observational study”. Please find our responses to the Editors’ and Reviewers’ comments shown in Bold in the following response letter.

Sincerely,

Nobuhiro Sato

 

Reviewers' comments:

6. Review Comments to the Author

Reviewer #1: The authors have addressed all my comments adequately.

The manuscript now looks nice.

I have no further comments.

Thank you for your thorough reviews and comments.

 

Reviewer #3: This study aimed to examine the predictors of difficulty of hospital acceptance of patients suspected to have cerebrovascular disease. It is unclear who and how to define those who were suspected to have cerebrovascular diseases? By the EMS personnel? 

We appreciate your suggestions. Hemiplegia, dysarthria, ataxia or severe headache were classified as cerebrovascular disease-related symptoms. The data were collected by EMS personnel, in cooperation with the physicians overseeing the patient’s care. We added information to the Methods section (Page 7, Para 108-109 and 111-112).

Difficulty of acceptance was defined by having ≥4 phone calls by the EMS personnel to hospitals until acceptance. Who are the one responding to the phone calls? Triage nurses? Hospital administrators? Physicians? 

We thank the reviewer for the careful review of the manuscript. The person who responds to the phone calls by EMS personnel depends on the hospital. While physicians respond to the phone calls in some hospitals, nurses or the other staff respond in the other hospitals. We have added a description about this in the Introduction (Page 5, Para 69-70).

Symptom presentation is a key factor to urge patients to call for help, same as healthcare professionals. Did the EMS personnel inform the one responding to the phone calls the symptoms of patients? Did they know that the patients are suspected to have cerebrovascular diseases? 

We appreciate your suggestions. Yes, EMS personnel informed medical staff responding to the phone calls about the symptoms which led EMS personnel to suspected cerebrovascular diseases according to EMS protocols.

Mean age of physicians is one of the factor associated with difficulty of hospital acceptance, was this variable refer to those who refused to admit the patients or those who finally admitted the patients? did these physicians involved in refusing to admit patients?

We appreciate reviewers' insightful comments. In this study, we analyzed hospital acceptance of each ambulance transport as opposed to acceptance of emergency admission to an inpatient ward from the emergency department. The variable mean age of physicians refers to the mean age of physicians in the area that each patient was transported from. Therefore, this is the mean age of physicians in the area who refused to accept the EMS transportation. We have added “in the area” to the Results and the Discussion (Page 12, Para 201, Page 17, Para 214 and 222-223, Page 20, Para 234, Page 21, Para 255, Page 23, Para 294).

Minimal grammatical errors, but the manuscript requires some formatting to make it more readable, such as the “≦” should put in front of 75 percentile, Table 1, the fourth column heading should be ≥4

We thank the reviewer for the careful review. We have corrected the typos.

---

## [Editor Report · Decision Letter 2]

29 Dec 2020

Factors associated with difficulty of hospital acceptance of patients suspected to have cerebrovascular diseases: a nationwide population-based observational study

PONE-D-20-03553R2

Dear Dr. Sato,

We’re pleased to inform you that your manuscript has been judged scientifically suitable for publication and will be formally accepted for publication once it meets all outstanding technical requirements.

Kind regards,

Ho Ting Wong, PhD

Academic Editor

PLOS ONE
---

## [Editor Report · Acceptance letter]

4 Jan 2021

PONE-D-20-03553R2 

Factors associated with difficulty of hospital acceptance of patients suspected to have cerebrovascular diseases: a nationwide population-based observational study 

Dear Dr. Sato:

I'm pleased to inform you that your manuscript has been deemed suitable for publication in PLOS ONE. Congratulations! Your manuscript is now with our production department. 

Kind regards, 

on behalf of

Dr. Ho Ting Wong 

Academic Editor

PLOS ONE